# The Spatial Patterns of the Crime Rate in London and Its Socio-Economic Influence Factors

Yunqi Zhou [1], Fengwei Wang [2,*] and Shijian Zhou [3]

1  School of Geographical Sciences, University of Bristol, Bristol BS8 1SS, UK; yunqi.zhou@bristol.ac.uk
2  State Key Laboratory of Marine Geology, Tongji University, Shanghai 200092, China
3  School of Software, Nanchang Hangkong University, Nanchang 330063, China; sjzhou@nchu.edu.cn
*  Correspondence: wangfw-foster@tongji.edu.cn

**Abstract:** This paper analyses the spatial trends and patterns of the crime rates in London and explores how socio-economic characteristics affect crime rates with consideration of the geographic context across London. The 2015 London Crime Statistics and Socio-economic Characteristics datasets were used. First, we investigated the spatial patterns of crime rates through exploratory spatial analysis at the ward level. In addition, both the ordinary least square (OLS) model and geographically weighted regression (GWR) model, which allow the effects of factors to vary in spatial scales, were adopted and compared to explore the potential spatially varying effect across London. The results showed that there exists obvious spatial clustering for the crime rate in central London. Both global and local Moran's I values indicated the spatial dependence of crime at the ward level. The GWR model performed better in explaining crime rates than the OLS model. Only two factors, namely, the percentage of children aged from 0 to 15 and employment rates, had significant spatial variability in London. The influences of the percentage of children aged 0 to 15 on crime rates are constantly negative over a spatial scale; however, employment rates positively affect crime rates in the north-western areas near the centre of London. Therefore, this paper focuses more on the spatial perspective, which fills the gap in traditional crime analysis, especially on the spatially varying influence of socio-economic status.

**Keywords:** spatial analysis; geographically weighted regression; crime rate

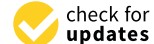



## 1. Introduction

Crime analysis is defined as involving both quantitative and qualitative methods that are applied to investigate and analyse the correlation among valuable and meaningful datasets, including crime incidents, disorder, internal police issues, and the quality of life of citizens (IACA 2014). There is no doubt that the key research point for crime analysis is to deeply study crimes and investigate their corresponding impact factors; many previous studies have quantitatively analysed crime events and explored their causes (Santos 2016). With the rapid development of geographical information science techniques and relevant computation skills, the spatial characteristics of crimes have attracted more attention to the study of how these victims and incidents aggregate and change on both temporal and spatial scales (Roth et al. 2010; Eck and Weisburd 1955). However, the majority of police departments lack advanced analytical tools, especially for some smaller regional levels, such as the ward level in the United Kingdom; therefore, they can only simply analyse the crime numbers and crime rates without much interpretation and explanation (Roth et al. 2010). Furthermore, most previous research publications used traditional regression methods, such as the ordinary least square (OLS) model, without considering variations in both temporal and spatial scales (Bellitto and Coccia 2018). It is worth noting that some scholars have considered the potential spatial patterns within crime rates, such as the usage of multi-level models to study the hierarchical geographical structures within datasets

(Thulin et al. 2021). Adeyemi et al. (2021) harnessed the Poisson version of generalized mixed models to incorporate the spatial-dependence effects and the higher-level, state-level heterogeneity effects of crimes to examine the geographic patterns of crimes. Tseloni et al. (2004) used multi-level negative binomial models to explore property crimes at the household and area levels. Tseloni et al. (2004) proposed that both household and area characteristics could explain the majority of variations in property crimes; however, there are still obvious between-area differences. The UK has a higher concentration of burglary victims than the US (Martinez et al. 2017). Other scholars focus on micro-units of analysis, such as the crime analysis at street segments conducted by Luo et al. (2022); they provide more evidence about how built environments and their interactions affect the occurrences of street crimes.

Based on the social structure and control on which offenders thrive, social disorganization theory (Bursik 1988; Sampson and Groves 1989) seeks to explain criminal activities (He et al. 2015). It is summarized and concluded that population migration, regional deprivation, and ethnic heterogeneity give rise to a higher crime rate according to a great deal of conducted research that aims to test the validity of the theory and explain the causes of crime events by harnessing multi-variable regression methods (Sampson 1985, 1987; Cahill and Mulligan 2003; Porter and Purser 2010; Bellitto and Coccia 2018). Social disorganization, in which a relatively low socio-economic status is important in contributing to criminal cases, including violence across neighbours by the reduction of collective efficacy and eroding social control, has been widely discussed (Lightowlers et al. 2023; Shaw and McKay 1942; Gruenewald et al. 2006; Mair et al. 2013). Shaw and McKay (1942) further explained that neighbourhood deprivation could be regarded as either contributing to offender motivations or strained social relations (Sampson et al. 1997), thereby leading to more crimes. Many studies have explored social disorganization theory and supported the arguments that deprived neighbours are associated with high risks of crime based on both individual-level and aggregated regional-level analyses (Lightowlers et al. 2023; Ciacci and Tagliafico 2020). More deprived neighbours polarize and exaggerate the differences among individuals and give rise to more crimes (Wilkinson 2004; Wilkinson et al. 2009; Lightowlers et al. 2023). For instance, Messer et al. (2006) discovered that women who live in economically deprived areas had more crime cases among their selected sample in Wake County, NC, US. Oyelade (2019) proposed that higher poverty is associated with higher crime rates in Nigeria after adopting time series crime rate data from 1990 to 2014. Oyelade (2019) explained that poverty may increase the circumstances of depression or mental illness, thus increasing crime. Livingston et al. (2014) proposed that the regional deprivation level affects crime because poverty could give rise to increased illegal activities aimed at obtaining goods to fulfil their needs. According to the British Crime Survey, household crime rates are 1.6 times higher in the most deprived quintile than in the least deprived quintile (Higgins et al. 2010). In addition to the neighbourhood economic deprivation level being regarded as the direct deprivation measure of socio-economic status, the unemployment rate could be regarded as one of the indirect representations of neighbourhood socio-economic status. Goh and Law (2023) provided evidence that higher employment rates are associated with low crime rates in Argentina, Brazil, and Chile, which is consistent with the empirical findings of Raphael and Winter-Ebmer (2001) showing that increased employment opportunities could deter potential offenders from committing crimes (Goh and Law 2023; Raphael and Winter-Ebmer 2001). The occurrence of frequent crime events has attracted increased attention in recent decades, as they not only affect the safety, health conditions, and quality of life of citizens at the individual level (Fazel et al. 2014) but also have a negative influence on societal development and stability (Kim et al. 2018). Currently, it is important to discover the underlying patterns of crime events and investigate the corresponding socio-economic factors affecting crime incidents. Comprehensive and in-depth crime analyses benefit police departments and the government with reduction and prevention strategies for criminal incidents, predicting future incidents, and related law enforcement matters (Roth et al. 2010; Santos 2016).

Some scholars have adopted spatial modelling methods to investigate spatial heterogeneity. Nezami and Khoramshahi (2016) adopted geographically weighted regression (GWR) methods to explore drug smuggling in the southern Khorasan Province; they indicated that more drug crimes were concentrated in the central province due to its geographic position and convenient road network. Andresen et al. (2021) examined how spatially varying unemployment affected criminal events in the city of Vancouver in British Columbia, Canada, by adopting GWR models. They indicated the existence of spatial heterogeneity regarding the associations between unemployment and criminal events, suggesting a spatially varying effect from place to place. Cahill and Mulligan (2007) adopted the GWR model to investigate urban violence in Portland, Oregon, and the US; single-person households had significant variation across space. However, few spatially varying effects have been explored in London.

Therefore, this paper aims to fill the gap in the geographical analysis of crime rates by considering the spatially varying influencing factors across London. Compared with other works that also adopted GWR models, this paper aims to focus on socio-economic status, including both the individual level (household income, education, employment, rent types) and neighbourhood level (rank of average score of deprivation, median house price). Furthermore, our analysis considered more demographic information, including the percentage of children and the percentage of immigrants not born in the UK. In addition, our work incorporates comparisons between different geographical scales. Compared to other scales, the ward level is a detailed geographical scale that could provide detailed spatial heterogeneity and, for policy-makers, more specific targeting of which areas need more attention.

The rest of this paper is organized as follows: Section 2 briefly introduces the study region and datasets used in this paper. Adopted methods, including exploratory spatial analysis and two regression analysis methods, are discussed in Section 3. Subsequently, spatial analysis and regression results are presented in Section 4. Then, the entire work is critically evaluated and discussed to point out the merits and shortcomings of the utilized analysis methods and indicate further improvements and future directions in Section 5. Finally, concluding remarks are given in Section 6.

## 2. Study Region and Datasets

London is one of the most internationally diverse cities in the world and so it was selected for this study. The datasets can be divided into two categories, specifically, Ward Profiles and Atlas and Borough Profiles and Atlas in the CSV format (Data.london.gov.uk 2022a, 2022b) and Statistical GIS Boundary Files (Data.london.gov.uk 2022c) for London at the ward and borough levels in the shape file format with the purpose of visualizing crime rate and socio-economic factors. Electoral wards are the key building blocks of UK administrative geography. They are the spatial units used to elect local government councillors (ONS 2023). The UK has 9196 electoral wards, and London has 624 wards. The population of each ward is approximately 5000 (ONS 2023). Compared with wards, boroughs have a relatively large geographic scale. The London boroughs are 33 local authority districts that compose the administrative area of London; each borough is governed by a London borough council (Data.london.gov.uk 2022a). The populations of the London boroughs were between 160,000 and 400,000 in 2014 (Data.london.gov.uk 2022b).

Our work focuses on the entire city of London, based on the London boundaries updated in 2014. In particular, an overview of demographical and relevant datasets, including information regarding the population, deprivation, health, education, crime, greenspace, welfare, labour market, housing, and transportation of the population at the ward level, is provided through Ward Profiles and Atlas, which combined 2011 Census data, GLA Population Projection data, London Ambulance Service data, Department for Work and Pension data, Metropolitan Police Service data, Transport for London data, and Greenspace information for Greater London (Data.london.gov.uk 2022a). Similarly, Borough Profiles and Atlas also provides such diversity of demographic and relevant

datasets of the population in each borough. The borough-level data merely aim to provide an overview of the distribution of crime events and rates in London. Simultaneously, the ward-level data are harnessed for further exploratory spatial analysis and regression analysis. The latest data regarding ward/borough profiles and atlases were updated in 2015. To maintain and achieve the consistency of the crime data and socio-economic characteristics and reduce bias, all used data are from the same source, which was recently updated in 2015.

## 3. Methodology

### 3.1. Exploratory Spatial Data Analysis

Exploratory data analysis aims to describe, explain, interpret, and visualize spatial distribution, as well as indicate spatial dependences, identify spatial outliers, including hot spots and cold spots, and investigate spatial association (Holt 2007; He et al. 2015; Tukey 1977). The exploratory spatial analysis is conducted first at the ward level to indicate the spatial auto-correlation of crime rates and identify the spatial outliers, which help us to comprehensively understand the spatial variation of crime rates over space, including the calculation of Global Moran's I, visualization of Local Moran's I, and spatial outlier display.

#### 3.1.1. Global Moran's I

Moran (1948) proposed Global Moran's I, one of the most widely utilized statistics to investigate spatial auto-correlation, which was referred to as the measurement of spatial dependence of random variables at different spatial locations (He et al. 2015; Zhang and Lin 2007). The equation of Global Moran's I is presented as follows (Griffith et al. 1991):

$$\text{I} = \frac{n}{\sum i \sum j W_{ij}} \frac{\sum i \sum j W_{ij} \left( X_i - \overline{X} \right) \left( X_j - \overline{X} \right)}{\sum i \left( X_i - \overline{X} \right)^2} \tag{1}$$

where $X$ is the studied variable, $W_{ij}$ is the spatial weight matrix, $X_i$ and $X_j$ are the observed values for features $i$ and $j$, and $\overline{X}$ is the mean of $X$ (Griffith et al. 1991).

In accordance with the formula, the linear correlation between an observed value and the spatially weighted average of neighbourhood values is displayed and interpreted by Global Moran's I value (Anselin et al. 2000); the neighbours are decided by the selected spatial weight matrix. In this case study, the spatial weight matrix is the continuity matrix in which regions that share boundaries are regarded as neighbours. The range of global Moran's I values is from −1 to 1, with positive global Moran's I values representing positive auto-correlation and negative values representing negative auto-correlation. The closer global Moran's I values are to −1 or 1, the stronger the spatial auto-correlation in the studied areas (Ratcliffe 2010). Furthermore, the z-score and $p$-value are harnessed in hypothesis testing to reject the null hypothesis of spatial independence (He et al. 2015). The 95% confidence interval is between −1.96 and 1.96 for the z-score. If the $p$-value is smaller than 0.05 and the z-score is not within the range spanning from −1.96 to 1.96, the spatial patterns cannot be regarded as consequences of the random spatial procedure and the spatial auto-correlation is statistically significant.

#### 3.1.2. Local Moran's I

Global Moran's I value is only capable of providing one summary value of the whole study area. However, the spatial dependences may change over spatial scales, thereby requiring local statistics to capture different spatial auto-correlations in different regions (Fotheringham and Brunsdon 1999). Anselin (1995) proposed the local Moran's I value to examine spatial clusters and outliers. The positive values represent areas whose neighbours are similar to themselves, including high and low crime rates; these areas are parts of the detected spatial clusters. Negative local Moran's I values indicate that these areas are spatial outliers that are different from their neighbours.

Both high–high and low–low clusters are represented by positive local Moran's I values, and both high–low and low–high outliers are revealed by negative local Moran's I values. Thus, it is meaningful to distinguish high–high clusters (hot spots) and low–low clusters (cold spots), as well as high–low outliers and low–high outliers, by combining the significance of local Moran's I values and locations of observations in the Moran scatter plot. It is noticeable that the hot spots and cold spots are based on the average crime rates and the average lag rather than representing absolute high and low crime rates.

*3.2. Regression Analysis*

Two regression models are harnessed and compared to investigate the influence of socio-economic factors on crime rates in London at the ward level with the same selected factors. The model explanation ability and accuracy are compared through statistical indicators, including adjusted R-squared.

3.2.1. Ordinary Least Squares (OLS) Regression Model

The ordinary least squares regression model is the most widely utilized regression model to estimate unknown variables in a linear regression model (Myers and Myers 1990; Leng et al. 2007). To obtain more valuable intercepts, both influence factors and dependent variables are centred on discovering the association between average socio-economic factors and average crime rates. The selections of the predicators are based on previous studies and research focused on socio-economic status. According to He et al. (2015), residential mobility, regional deprivation, and ethnic heterogeneity are associated with crime rates; their corresponding factors are summarized in Table 1. The rank of the average score of deprivation is based on the index of deprivation (IMD). It has been discussed previously that London has 624 wards and each ward has its corresponding rank regarding the average deprivation score (1 being the most- and 624 the least-deprived district in London). The Index of Multiple Deprivation is based on 39 separate indicators across 7 domains of deprivation to provide a comprehensive measure of relative deprivation in small areas in 2010 (IMD 2010). In addition to regional deprivation, individual wealth could also be considered in the regression. Additionally, Lochner and Moretti (2004) pointed out that education level is also associated with crime rates and that the higher the education level is, the lower the crime rates. Renting in the UK can be divided into private renting and social renting (UK Census 2011). Private renting refers to renting through a private landlord or letting agent and social renting refers to renting through a local council or housing association, which is less costly (UK Census 2011). Social-rented housing is allocated by councils and local councils have their own policies on who is eligible for social housing and who has priority (UK Census 2011). Age is also considered when investigating whether there are more crimes at young ages in London. The collinearity problems among factors are evaluated and checked by the variance inflation factor (VIF) with a threshold of 10 (O'brien 2007). A total of three variables, namely, median house price, percentage of English as the first language of no one in the household, and percentage with no qualifications, whose VIF values over 10, have been removed from the regression model.

**Table 1.** The selected factors in the OLS model and their corresponding themes.

| Themes | Influence Factors |
|---|---|
| Individual wealth | Employment Rate<br>Median Household Income |
| Regional deprivation | Median House Price<br>Rank of Average Score of Deprivation |
| Ethnic | Percentage Not Born in UK<br>Percentage of English as the First Language of no one in household |
| Residential mobility | Percentage Households Social Rented<br>Percentage Households Private Rented |
| Education | Percentage with No Qualifications<br>Percentage with Level 4 Qualifications and above |
| Transportation | Transport Accessibility Score |
| Age | Percentage All Children aged 0 to 15 |

### 3.2.2. Geographically Weighted Regression Model

The geographically weighted regression model extends the traditional regression method by allowing the effects of factors to vary over space; it is capable of interpreting the spatial variations in the coefficient estimations of different independent variables based on the Gaussian decay of the inverse distance weighted approach rather than utilizing global coefficient estimations across the whole study region (Brunsdon et al. 1998; Gollini et al. 2013). The selected socio-economic factors are the same as the OLS model, for comparison. The bandwidth is manipulated through the cross-validation optimization method (Cho et al. 2010). Subsequently, the Monte Carlo significance test is harnessed to evaluate the significance of the coefficient variability of the factors (Gollini et al. 2013).

## 4. Results and Analysis

### 4.1. Overall Understanding of Crime Rates

First, the quantity of criminal events at the borough level is displayed in Figure 1, regarding the time period of 2014–2015. In contrast, it is more accurate to utilize crime rates in further analysis, rather than the number of criminal events, to better represent the actual situation of crimes in different regions. The crime data are derived from the Metropolitan Police Service and the calculated crime rates are per thousand of the population (we divided the number of crime events by the entire population in each borough and then multiplied by 1000 to obtain the crime rate). The crime rate at the borough level is exhibited in Figure 2, with the size of the point indicating the population who lives in the borough. It is worth noting that the City of London borough is not under the charge of the Metropolitan Police Service and the analysis of the crime rate in London invariably excludes the City of London borough (Met.police.uk 2022). Furthermore, according to Figures 1 and 2, it can be observed that the population in the City of London is much smaller compared with other boroughs. Although the quantity of crime events is not so high, the calculated crime rate is much higher than in other boroughs. Therefore, it is better to exclude the City of London to conduct a more reasonable comparison and analysis of the crime rates in London. Figure 3 shows the crime rate with the corresponding population at the borough level, without the City of London borough. It is noticeable that the scopes of the crime rate and population are more reasonable, with more evident and instinctual differences in crime rates across London.

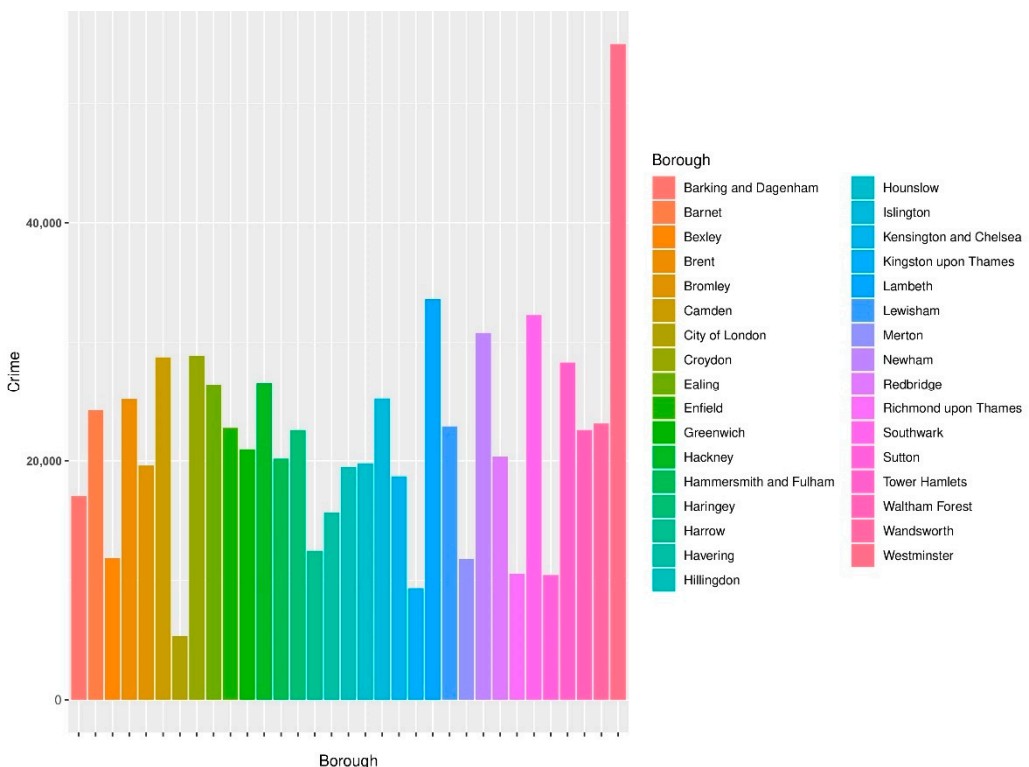

**Figure 1.** Quantity of criminal events at the borough level in London (2014–2015).

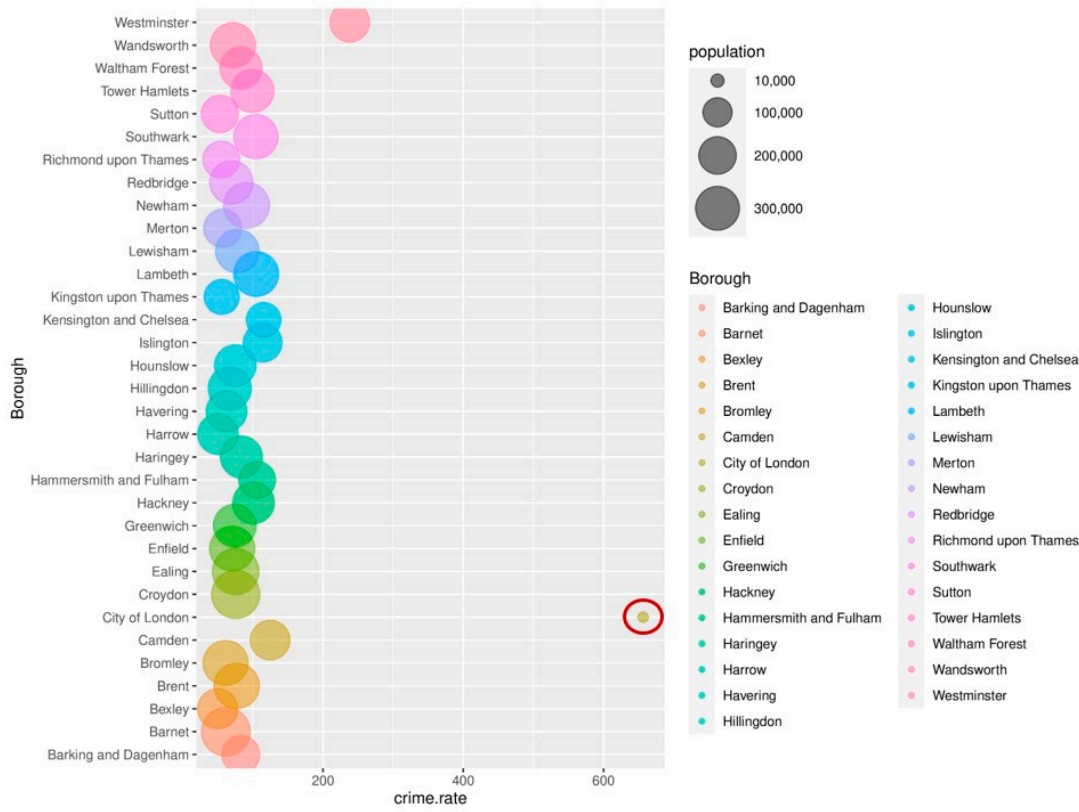

**Figure 2.** Crime rate with population at the borough level in London (2014–2015).

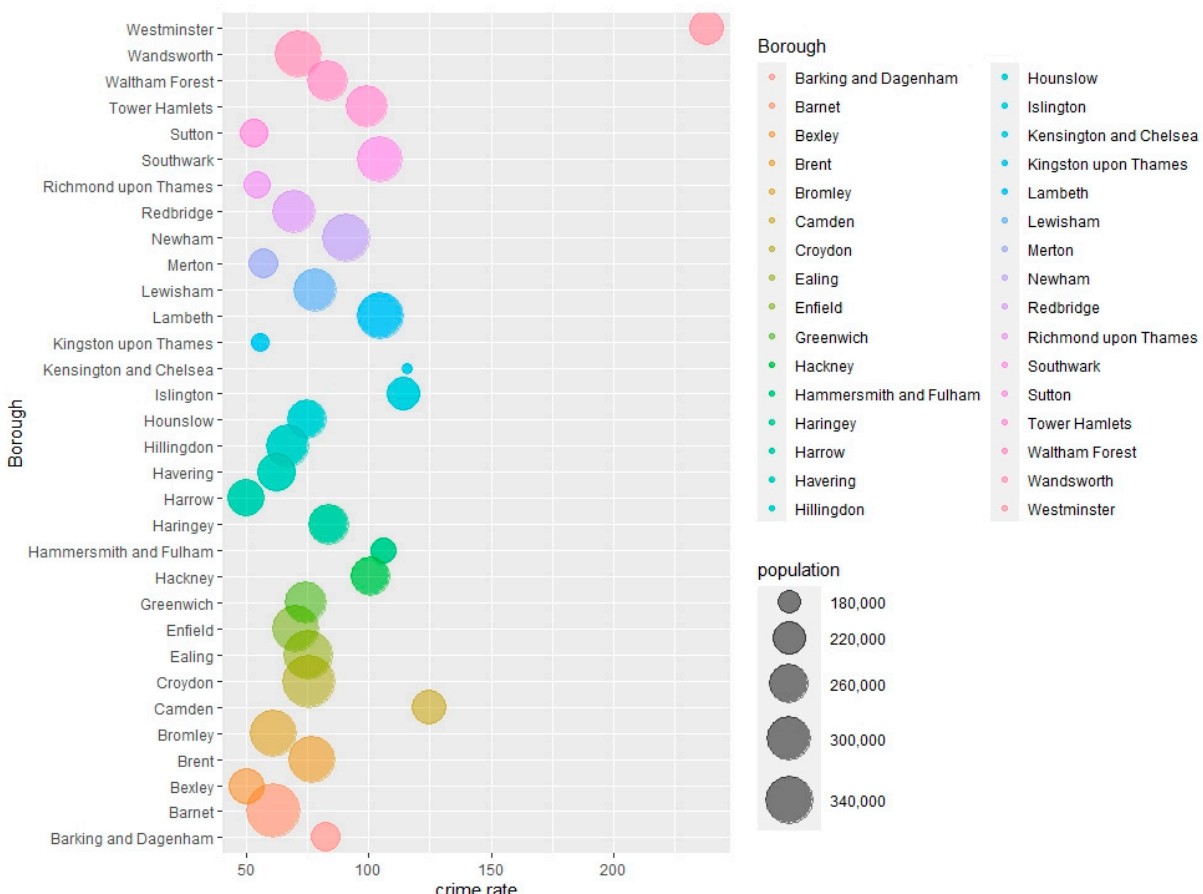

**Figure 3.** Crime rate with population at the borough level in London, without the City of London (2014–2015).

Based on the histogram plots in Figure 4, the crime rates are not uniformly distributed with positive skewness and there are relatively low crime rate clusters at both the borough and ward levels. Therefore, the natural break classification method is harnessed in producing the choropleth map to maximize the differences between different groups of crime rates and better capture the statistical characteristics of crime rate distribution (ESRI 2019). Simultaneously, the utilized shading scheme is that the higher the crime rate is, the darker the colour; this is implemented to visualize the spatial distribution and patterns of crime rates in London.

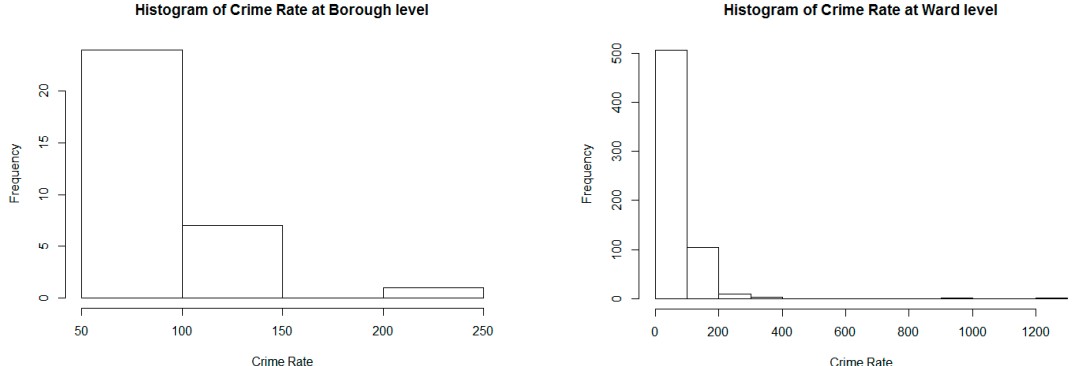

**Figure 4.** Histogram of crime rates at the borough level and ward level in London, without the City of London (2014–2015).

According to Figure 5, which represents the crime rates at different levels, it is obvious that there exist clustering patterns of crime rates with high/low levels at borough/ward levels in London. The high crime rates are clustered and aggregated in the centre of London, while the low crime rates are clustered in the marginal areas, including the north-western, north-eastern, and southern areas, indicating the existence of a spatial auto-correlation of crime rates over space. Furthermore, the differences between crime rates at the ward level are much greater than those at the borough level, illustrating that the spatial variation in crime rates, which is referred to as spatial heterogeneity, is very noticeable and cannot be neglected in the ward-level analysis.

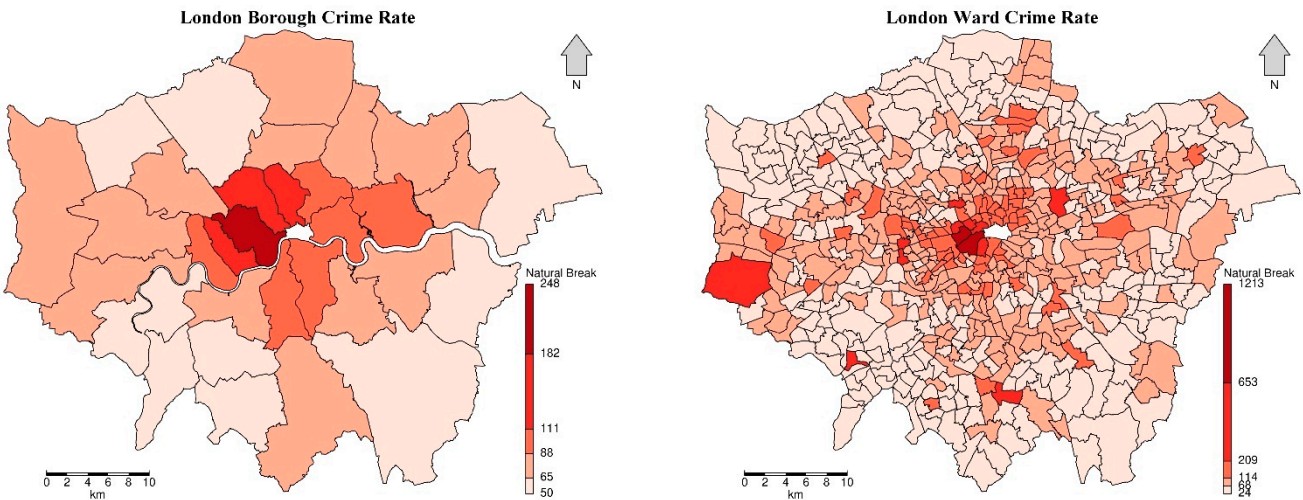

**Figure 5.** Histogram of crime rate at borough/ward level in London, without the City of London (2014–2015).

### 4.2. Exploratory Spatial Analysis Results

The global Moran's I values and relevant statistical indicators are shown in Table 2. The *p*-value is close to 0, and the z-score value is much greater than 1.96, rejecting the null hypothesis of spatial independence and indicating that the global Moran's I value is statistically significant. A positive global Moran's I value (0.378) indicates that there exist spatial clusters of crime rates in London at the ward level.

**Table 2.** Global Moran's Summary.

| Moran's Index | z-Score | *p*-Value |
| --- | --- | --- |
| 0.378 | 18.299 | $<2.2 \times 10^{-16}$ |

Figure 6 shows the local Moran's I values of the crime rate; areas with statistically significant local Moran's I values are highlighted with a black border. The statistically significant regions are clustered in the centre of London with extremely high local Moran's values, which is similar to the distribution map of the crime rates. The distribution of regions with negative Local Moran's I values is very sporadic; some of them surround regions with extremely high Local Moran's I values. A positive local Moran's I indicates that clusters exist. However, the types of clusters (high–high and low–low) cannot be determined through local Moran's I values.

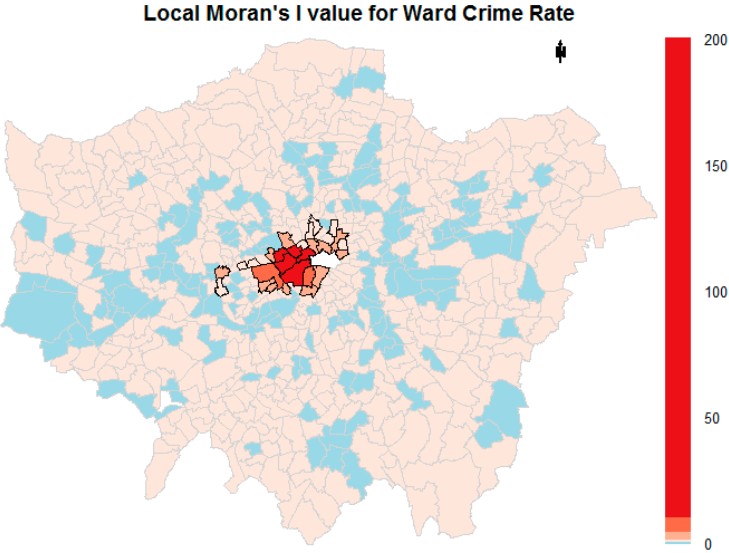

**Figure 6.** Local Moran's I values of crime rates and areas (the statistically significant values are highlighted with a black border).

Spatial lags of crime rates are the spatially weighted average crime rates of neighbourhoods of the regions. Figure 7 shows the Moran scatter plot between crime rates and their corresponding spatial lags of crime rates. According to Figure 7, there are four districts that can explain spatial auto-correlation with different spatial patterns (Anselin 1995). The majority of the observations are clustered in the low–low pattern.

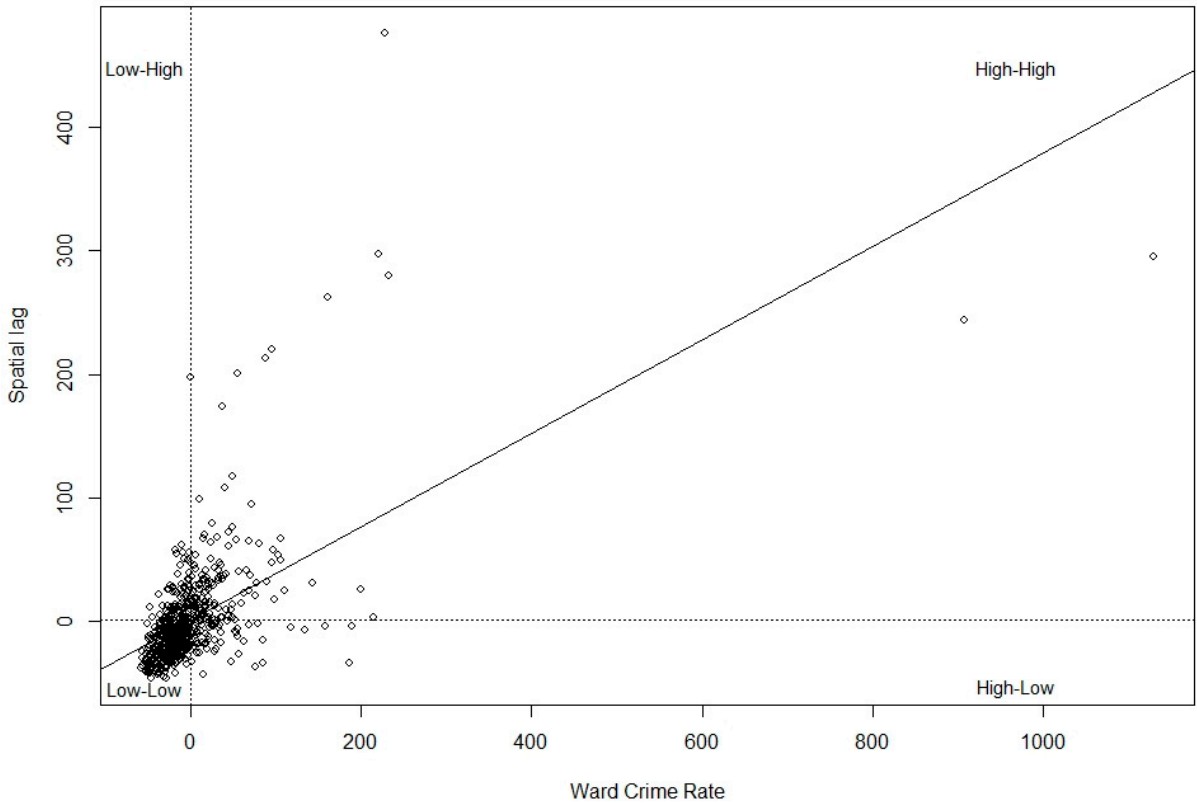

**Figure 7.** Moran Scatter plot of crime rate at the ward level.

The hot and cold spots of crime rates and regions are highlighted with a black border, which could be considered statistically significant with a *p*-value smaller than 0.5, as shown in Figure 8. The significant regions are high–high patterns, being clustered in the centre of London. The low–low patterns cover the majority of areas in London, which is consistent with the results of the Moran scatter plot. The crime rates in the Westminster districts are extremely high, which exaggerates the mean crime rates and spatial lags. Therefore, the majority of areas are regarded as low–low patterns due to the hot and cold spots, representing the relative values compared with the mean rather than absolute high or low values of crime rates.

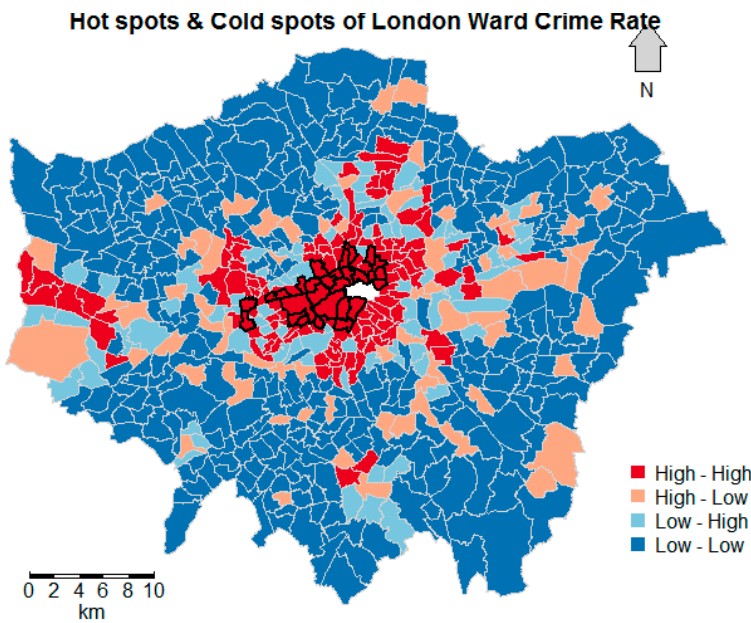

**Figure 8.** Hot spots and cold spots of crime rates and areas highlighted with black borders are statistically significant.

*4.3. Regression Analysis*

4.3.1. OLS Model Results

After the calculation and examination of VIF, the selected independent variables are shown in Table 3. Among all influence factors, the most substantial influence on crime rates is the transport accessibility score; the higher the transport accessibility scores are, the higher the crime rates. A higher percentage of children aged from 0 to 15 is negatively associated with crime rates, illustrating that child crime cases are not the primary type of criminal event in London. It is not surprising that a higher percentage of people with level 4 and above qualifications leads to lower crime rates. In terms of residential mobility, both the percentage of socially-rented households and the percentage of privately-rented households are positively associated with crime rates. Higher median household income is concomitant with higher crime rates. Three independent variables, namely, the percentage of those not born in the UK, employment rate, and rank of the average score of deprivation, are not statistically significantly related to crime rates.

**Table 3.** Coefficient, standard errors, *p*-values, and adjusted R-square of the OLS model.

| Variables | OLS Model | |
|---|---|---|
| | **Coefficient Estimations** | **Standard Error** |
| Intercept | $9.86 \times 10^{-15}$ | 2.3710 |
| Percentage All Children aged 0 to 15 | −3.1620 *** | 0.9263 |
| Percentage Not Born in UK | 0.1447 | 0.3487 |
| Employment Rate | 0.5799 | 0.7513 |
| Median Household Income | $3.449 \times 10^{-3}$ *** | 0.7941 |
| Transport Accessibility score | 16.7500 *** | 3.4490 |
| Percentage with Level 4 Qualifications and above | −3.3680 *** | 0.5356 |
| Rank of average Score of Deprivation | 0.0367 | 0.0355 |
| Percentage Households Social Rented | 1.4560 *** | 0.4168 |
| Percentage Households Private Rented | 2.4910 *** | 0.6381 |
| Adjusted $R^2$ | 0.3007 | |

*** $p < 0.001$.

According to Figure 9, the variance of residuals of the OLS model is not constant, which displays evidence of heteroscedasticity. The 0 line (with zero residuals) is the reference line, and the red line is a smoothed curve fit between the residuals and the fitted values. When the linear regression model fits well, the residuals are more or less randomly distributed around the 0 line to indicate that the variances of the errors are equal. However, the assumptions of independent and identically distributed errors are violated because residuals are not randomly distributed around the 0 line, but when fitted values are larger, the residuals are relatively more negative. This may give rise to underestimated standard errors and unreliable t- and *p*-values of factors derived from the OLS model.

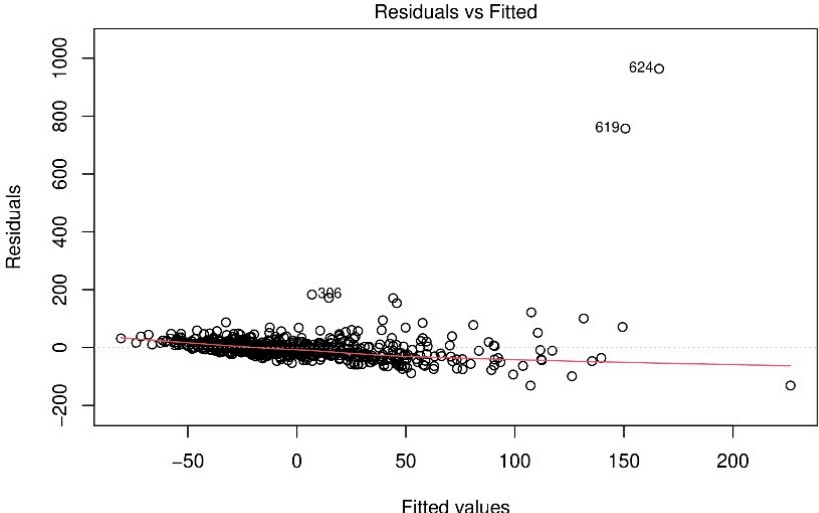

**Figure 9.** Residuals vs. Fitted (Predicted) values of crime rates.

The spatial patterns of the residuals of the OLS model are evident and apparent; they are visualized and interpreted in Figure 10. There also exist spatial clusters of residuals, including high–high patterns clustered in the centre of London and low–low patterns located in the areas adjacent to the centre of London.

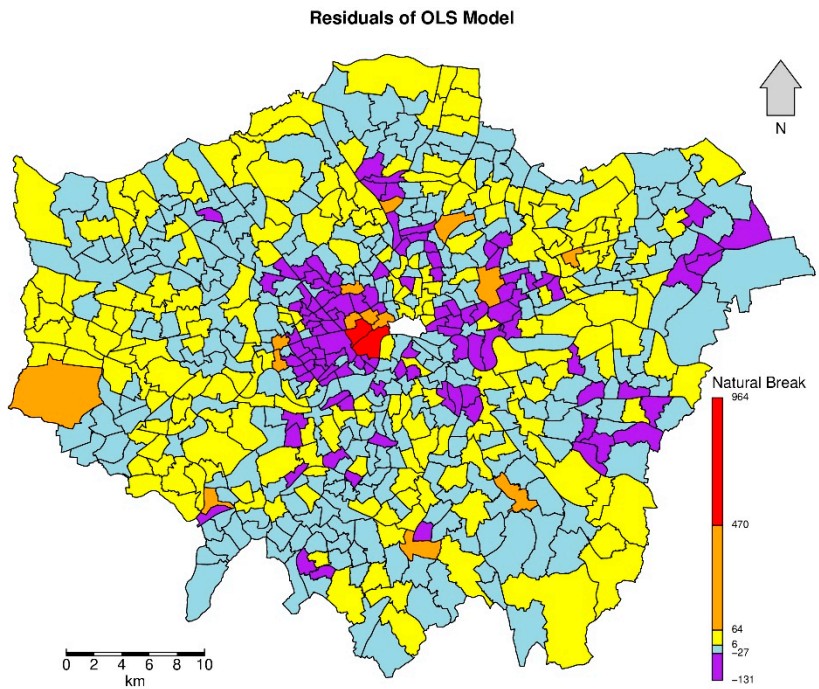

**Figure 10.** Residuals of the OLS Model.

### 4.3.2. GWR Model Results

The variations of the coefficient estimations of independent variables, *p*-values (significance of variability of factors), and adjusted R-squared are exhibited in Table 4. Only two factors, namely, the percentage of children aged from 0 to 15 and the employment rates, have significant spatial variability over space in London. There is only one coefficient estimation for each prediction in the OLS model, which is close to the median estimation in the GWR model. Although the global influence of the employment rate on crime rates is positive, in certain areas, the positive impact of the employment rate is strong and substantial. In contrast, the impact may even be negative in other areas. Therefore, the coefficient estimations in OLS may underestimate and overestimate the influence of the two factors, including the percentage of children aged from 0 to 15 and the employment rates in different regions.

**Table 4.** Variations of coefficient estimations and adjusted R-squared of the GWR model.

| Index | Min. | 1st Qu. | Median | 3rd Qu. | Max | *p* Values |
|---|---|---|---|---|---|---|
| Intercept | −5.234 | −2.387 | −0.689 | 3.111 | 9.357 | 0.017 |
| Percentage All Children aged 0 to 15 | −10.050 | −6.146 | −4.365 | −3.162 | −2.043 | 0.002 |
| Percentage Not Born in UK | −1.131 | −0.219 | −0.068 | 0.054 | 0.4594 | 0.518 |
| Employment rate | −0.165 | 0.560 | 1.206 | 2.246 | 4.987 | 0.048 |
| Median Household Income | 0.0032 | 0.0037 | 0.0041 | 0.0049 | 0.0064 | 0.952 |
| Transport Accessibility score | 14.360 | 17.630 | 18.890 | 20.320 | 22.750 | 0.724 |
| Percentage with Level 4 qualifications and above | −8.641 | −5.360 | −4.163 | −3.589 | −2.881 | 0.426 |
| Rank of average score of deprivation | 0.0024 | 0.0181 | 0.0344 | 0.0059 | 0.0850 | 0.540 |
| Percentage Households Social Rented | 1.105 | 1.357 | 1.668 | 1.910 | 2.534 | 0.737 |
| Percentage Households Private Rented | 1.973 | 2.417 | 2.796 | 3.076 | 4.3840 | 0.752 |
| Adjusted R$^2$ | 0.3587 | | | | | |

The coefficient variations of the other seven factors, namely, percentage not born in the UK, median household income, transport accessibility score, percentage of the population with Level 4 and above qualifications, rank of the average score of deprivation, percentage of rented social households, and percentage of privately rented households, are insignificant, indicating that the influence of these factors could be treated as the same values across the study region. Most importantly, the adjusted R-square of GWR is 0.3587, which is higher than that of the OLS model.

Distribution maps of the coefficients of these two factors with significant variations are exhibited in Figure 11. Regions whose coefficients are statistically significant at the 95% level are highlighted with black boundaries. The negative influences of the percentage of children aged from 0 to 15 on crime rates are significant throughout the entire study region. In terms of employment rates, regions with significant positive impacts are located in the north-western areas near the centre of London. The influence of employment rates on the crime rate in the eastern parts of London is negative. However, their corresponding estimated coefficient is not significant. Consequently, employment rates affect crime rates positively in the north-western areas near the centre of London, with spatial variability.

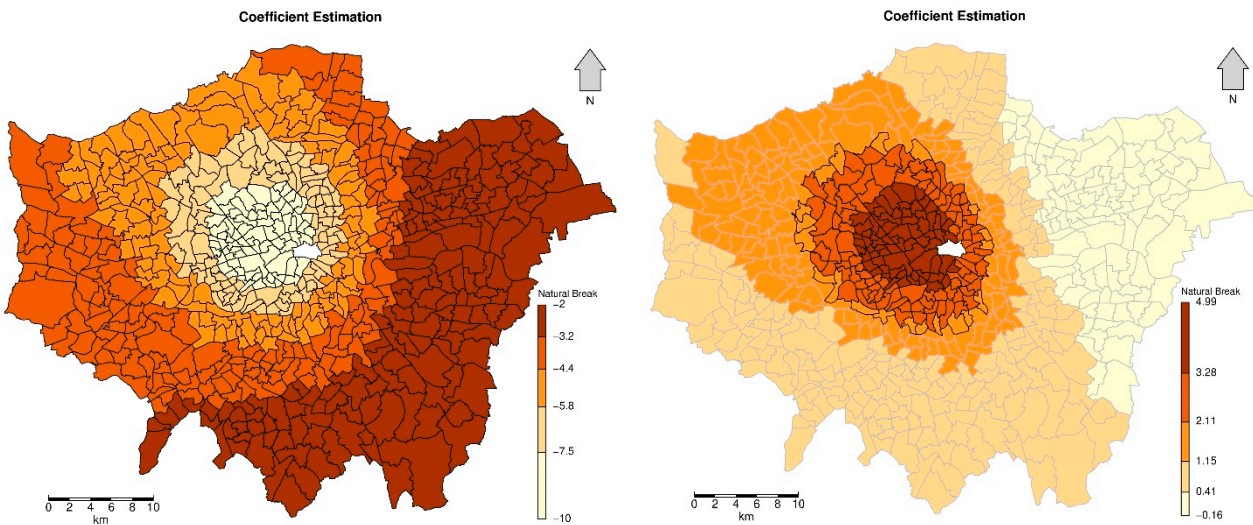

**Figure 11.** Coefficient estimation of the percentage of children aged from 0 to 15 (**Left**) and employment rate (**Right**) (Areas highlighted with black borders are statistically significant).

## 5. Discussion

### 5.1. Comparisons between OLS and GWR Models

In accordance with both the ward-level and borough-level crime rate maps in London, the spatial heterogeneity (variations in crime rates over space) and spatial dependence (hot spots in the centre and cold spots in the marginal areas) are apparent and noticeable. Additionally, the spatial clustering patterns could also be observed from the residuals of the OLS model and the non-constant variances of residuals violate the assumption of independent and identically distributed residuals. The existence of spatial auto-correlations in residuals of the OLS model may overestimate the significance of independent variables and increase the instability of measurement effects (Harris 2016). Therefore, it is beneficial to harness the regression model considering spatial dependence to improve the prediction accuracy and reduce bias.

The geographically weighted regression model performs better than the OLS model according to a higher adjusted R-square value. The explanation ability and prediction accuracy of the OLS model is reduced and restricted due to the "one size fits all" coefficient estimations, which neglect the variations of independent variables over space. Spatial non-stationary effects and spatial dependence are considered in the GWR model with a better investigation of the spatial heterogeneity through the harness of multiple coefficient

estimations of factors, which improves the data fitting and model accuracy compared with the OLS model.

*5.2. Policy Implications*

The results from the GWR model indicate particularly relevant policy implications, as there should be increased recognition of the importance of the context of crimes, particularly the location and geography of criminal events (Andresen et al. 2021). The emphasis on local differences in the context of crime can bring benefits to specific policy strategies and avoid one-size-fits-all policies. The GWR models present the non-stationary influence between socio-economic status and crime across London, thus highlighting where more attention is needed. For example, coefficient estimations for the proportion of children differ significantly between central and eastern London. Global models, such as OLS models, which provide only constant coefficient estimations, would suggest a global policy for London as a whole. However, global policy may be highly effective in some areas, while resources may be wasted in others. Andresen et al. (2021) pointed out that the global effect may be undetected, leading to the assumption that the policy is ineffective when in fact it may be highly effective in a small number of places that could benefit greatly from such a policy (Andresen et al. 2021). Our analysis suggests that London should not be considered as a whole to propose global policies but rather that policies should be enacted on a place-by-place basis, depending on the differential spatial impacts of the predictors. The division of London is also crucial in policy-making, for example, how many boroughs or districts should be considered a part of central London. Boroughs can provide rough estimates, but wards are more detailed and allow for maintaining more detail on spatially varying effects, which is one of the reasons why this analysis is conducted at the ward level. Furthermore, even though the signs of associations between the employment rate and crime rate switched from negative to positive, areas with negative relationships were not statistically significant. Thus, the discussions on spatial heterogeneity should therefore be based on both changes in signs and statistically significant tests.

*5.3. Further Improvement and Future Research Directions*

It is worth noting that the City of London is excluded from this case study in order to address its high crime rate due to its smaller number of residents. In the current calculation method of crime rate, only residents are included without consideration of the floating population, such as travellers. Suppose the floating population could be estimated and each ward could be assigned with the indicators of population mobility. In that case, the crime rate of the City of London may be calculated using the new approach and could be considered in the whole analysis procedure.

In addition to the GWR model, there are other spatial models, such as the spatial error model, which harnesses statistical nuisance to consider spatial auto-correlations, and the spatially Y lag model, which considers the spatial spillover influence on the crime rate at each location to handle spatial dependences. Furthermore, the multi-level modelling approach could also be utilized to allow the spatial variations of the influence of factors at different regional levels. Therefore, it is beneficial to conduct regression analysis based on more approaches, investigate the advantages and disadvantages of different models, and seek a model which could better interpret the impact of socio-economic characteristics on crime rates with better data fitting.

Crimes could be classified into nine types: theft and handling, violence against a person, burglary, criminal damage, drugs, robbery, sexual offences, other notifiable offences, and fraud and forgery (Met.police.uk 2022). Therefore, future research directions could focus on the socio-economic influences on different types of crimes. Additionally, the UK census has only three age groups, including 0–15 years old, 16–64 years old, and older than 64 years old. Thus, we only considered the age group between 0 and 15 years old. Due to the data limitation, it is difficult to obtain family structure information (e.g., single-parent/mother households) in London. Thus, the variables related to family structures

were not considered in our analysis. We hope to conduct further analysis in the future to consider the family structure and 15–30 years old populations when data are available. The overall analysis is a cross-section due to the data restriction in the Ward Profiles and Atlas. It is beneficial to conduct a longitudinal study of crime rates and discover temporal changes during recent years. A spatiotemporal analysis of crime rates in further research with a combination of socio-economic characteristics can comprehensively understand the tendency and changes in crime rates and the influence of associated factors over space and time (Fotheringham et al. 2015).

## 6. Conclusions

This paper focuses on analyzing the spatial patterns of crime rates and how to utilize different regression models to investigate the influence of socio-economic characteristics on crime rates in London. According to the ward and borough levels of crime rates, it is noticeable that there are spatial clusters in London's centre. Subsequently, exploratory spatial analysis is conducted based on Global Moran's I and Local Moran's I values. Both global and local statistical indicators reveal the spatial dependence of crime rates at the ward level. Afterwards, two regression analysis methods are compared, including the ordinary least square regression and geographically weighted regression methods. The heteroskedasticity of residuals derived from the OLS model violates the assumption of independent and identically distributed errors; the obtained standard errors and *p*-values are not reliable. The GWR method performs better with a higher adjusted R-square because it allows the influence of factors to vary across space. Finally, the comparisons of the two models have been critically discussed and further improvements and future directions are discussed. This work focuses more on the spatial perspective, which fills the gap in traditional crime analysis.

**Author Contributions:** Conceptualization, Y.Z. and F.W.; methodology, Y.Z.; validation, Y.Z., F.W. and S.Z.; formal analysis, Y.Z.; data curation, Y.Z.; writing-original draft preparation, Y.Z. and F.W.; writing-review and editing, F.W. and S.Z.; supervision, S.Z.; project administration, S.Z.; funding acquisition, S.Z. All authors have read and agreed to the published version of the manuscript.

**Funding:** This research was funded by the National Natural Science Foundation of China (42064001).

**Institutional Review Board Statement:** Not applicable.

**Informed Consent Statement:** Not applicable.

**Data Availability Statement:** Not applicable.

**Acknowledgments:** The authors are grateful to the editor Iva Milutinovic and four anonymous reviewers for their comprehensive and insightful comments, which have led to an improved presentation of the manuscript.

**Conflicts of Interest:** The authors declare no conflict of interest.

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
