# Peer review of "The Spatial Patterns of the Crime Rate in London and Its Socio-Economic Influence Factors"

_socsci, doi:10.3390/socsci12060340_

Round 1

Reviewer 1 Report

Front-end

-There are many more scholars beyond Adeyemi et al. who have used poisson multi-level models, including negative binomial models at multiple levels, as well as looking at spatial analysis at the street segment and micro unit of analysis. You should highlight others and the relevant literature.

-Also, I disagree with the last sentence of the first paragraph. Many studies have considered the varying spatial effect of factors on crime. With that said, how does your study differ from other studies? Or how does it push literature forward? I see at the beginning of the third paragraph you state that this paper fills the gap by “consideration of the spatial varying influence…” one or two more sentences are needed to explain why.

-Social disorganization was also extended in the 1980s/1990s with the advent of concentrated disadvantage (see William Julius Wilson’s work).

Data/Methods

-What is your unit of analysis and what is it the smallest unit of analysis? International audiences may not know if a ward vs. borough the smaller/larger unit is.

-Are you examining the entire city of London? Just the jurisdiction the police patrol? Further explanation of how you truncate your study area is needed.

-You repeat sentence that starts on line 85 into line 90.

- Line 146 “The selections of predicators are based on previous studies and personal perspectives.” What do you mean by personal perspectives? Not necessarily empirical.

-What factors had high VIF scores? What factors were removed?

-How does “individual wealth” differ (statistically) from “regional deprivation”? Why would you not combine those measures into one factor to measure “concentrated disadvantage”? Your measure of poverty/deprivation may be stronger if combining these together. Additionally, how did you create these measures? Through Cronbach Alpha scores? Factor loadings? More detail about the measures is needed.

-What about family structure (e.g., single-parent/mother households)? That is often a criminogenic factor that is combined to measure concentrated disadvantage.

-Did you take the number of crime events and divide them by the population of each Borough, and then multiply by 1,000? That needs to be more clearly stated.

-Why did you control for those who age 0-15 instead of saying, 15-30? Age-crime curve literature suggests that criminal propensity increases in the late teens. Further, it is rare for a 0–15-year-olds to commit a crime.

-You mention Moran’s I and the possibility of spatial dependence – how did you control for that in your models? Did you use a spatial lag?

Discussion

-Based on the title of the article and subsequent abstract, a deeper discussion is needed regarding the SES variables used to examine spatial pattering. Of the significant (and non-significant) findings, what implications does this have for the field/policy?

-Limitations of the 0- to 15-year-old measure need to be highlighted.

-More discussion in general about policy/practice is needed.

-in sum, we have been able to establish that crime clusters in certain areas, and that high-crime areas suffer from social ills. How does this study add to the literature? It seems that this is more of a methodological/analytical piece. It should be framed as such.

Miscellaneous comments

-The manuscript needs a thorough re-read/edit for grammar errors, word choice, syntax. For example, on line 29 it should read “With the rapid development…” instead of the “With the rapidly development…” Also, there are times when crime rate (singular) should be used while crime rates (plural) should be used.

-Put the links of the data in parentheses, makes it confusing to the reader.

See above comments.

Author Response

Thank you for offering the opportunity to revise our paper. First of all, the editor and reviewers are greatly acknowledged for their constructive comments, which are all implemented in the revised version. And all the revisions are shown with the red letter in the revised manuscript. We have provided point-by-point answers to the reviewers’ comments in the attachment file.

Reviewer 2 Report

Dear Authors, I enjoyed reading your article, which is a well-written piece of research on the geographical distribution of crime in relation to the selected criminological/social-economic factors. I recommend the article for publication as such. 

I am not a native English-speaking person. The article reads well and is comprehensive. 

Author Response

Thanks very much for your comments. We have carefully checked the English language of whole manuscript.

Reviewer 3 Report

Some  editing for English-language usage is needed.

Author Response

(The authors gave the same response as above.)

Reviewer 4 Report

This article addresses an interesting and relevant topic within social sciences and criminological research. The article is methodologically and empirically robust and well thought through. However, it lacks theoretical background analysis, which is visible then in the discussion of the data. This means that the article does not provide a significant or sufficient discussion of the literature on crime trends and how these might be affected by the socioeconomic characteristics of geographical areas. In the introduction, the authors very briefly pass through this, but no actual identification or discussion of this extensive literature is provided to frame the research conducted. This affects the discussion of the data later in the article, where the authors do not discuss the results against the literature. Also, no implications of this analysis are referred to public policy.

This is a relevant, well-written and high-quality empirical study, which lacks background contextualization through scientific literature published in this area of research and an in-depth discussion. Therefore, I would recommend the authors add a literature review section (or expand on this in the introduction) and then develop accordingly the discussion of the data against this literature and how it might impact public policies.

The article is very well-written, although it needs to be proofread as some sentences are missing a clear structure.

Author Response

Thank you for offering the opportunity to revise our paper. First of all, the editor and reviewers are greatly acknowledged for their constructive comments, which are all implemented in the revised version. And all the revisions are shown with the red letter in the revised manuscript. 

Reply to the reviewer’s comments on the manuscript entitled "The spatial patterns of crime rate in London and its socio-economic influence factors"

Reply to the editor and reviewers:

Thank you for offering the opportunity to revise and resubmit our paper. First of all, the editor and reviewers are greatly acknowledged for their constructive comments, which are all implemented in the revised version. And all the revisions are shown with the red letter in the revised manuscript. Below we provide point-by-point answers to the reviewers’ comments.

Reviewer #4

This article addresses an interesting and relevant topic within social sciences and criminological research. The article is methodologically and empirically robust and well thought through. However, it lacks theoretical background analysis, which is visible then in the discussion of the data. This means that the article does not provide a significant or sufficient discussion of the literature on crime trends and how these might be affected by the socioeconomic characteristics of geographical areas. In the introduction, the authors very briefly pass through this, but no actual identification or discussion of this extensive literature is provided to frame the research conducted. This affects the discussion of the data later in the article, where the authors do not discuss the results against the literature. Also, no implications of this analysis are referred to public policy.

Thank you so much for indicating the necessity for adding more literature reviews about crime trends and how these might be affected by socio-economic characteristics of geographical areas. We have added the following content in the introduction part.

Social disorganization, in which relatively low socioeconomic status is important in contributing to crime cases, including violence across neighbours by the reduction of collective efficacy and eroding social control, has been widely discussed (Lightowlers et al., 2023; Shaw and Mckay, 1942; Gruenewald et al., 2006; Mair et al., 2013). Shaw and Mckay (1942) further explained that neighbourhood deprivation could be regarded as either offender motivations or strained social relations (Sampson et al., 1997), thereby leading to more crimes. Many studies have explored social disorganization theory and supported the arguments that deprived neighbours are associated with high risks of crime based on both individual-level and aggregated regional-level analyses (Lightowlers et al., 2023; Ciacci and Tagliafico, 2020). More deprived neighbours polarize and exaggerate the differences among individuals and give rise to more crimes (Wilkinson, 2004; Wilkinson and Pickett, 2009; Lightowlers et al., 2023). For instance, Messer et al. (2006) discovered that women who live in economically deprived areas had more crime cases among their selected sample in Wake County, NC, US. Oyelade (2019) proposed that higher poverty is associated with higher crime rates in Nigeria by adopting time series crime rate data from 1990 to 2014. Oyelade (2019) explained that poverty may increase the circumstances of depression or mental illness, thus increasing crime. Livingston et al. (2014) proposed that the regional deprivation level affects crime because poverty could give rise to increased illegal activities aimed at obtaining goods to fulfil their needs. According to the British Crime Survey, household crime rates are 1.6 times higher in the most deprived quintile than in the least deprived quintile (Higgins et al., 2010). In addition to the neighbourhood economic deprivation level as the direct deprivation measure of socioeconomic status, the unemployment rate could be regarded as one of the indirect representations of neighbourhood socioeconomic status. Goh et al. (2023) provided evidence that higher employment rates are associated with low crime rates in Argentina, Brazil and Chile, which is consistent with the empirical findings of Raphael and Winter-Ebmer (2001) that increased employment opportunities could deter potential offenders from committing crime (Goh et al., 2023; Raphael and Winter-Ebmer, 2001).

This is a relevant, well-written and high-quality empirical study, which lacks background contextualization through scientific literature published in this area of research and an in-depth discussion. Therefore, I would recommend the authors add a literature review section (or expand on this in the introduction) and then develop accordingly the discussion of the data against this literature and how it might impact public policies.

Thank you so much for indicating the necessity for adding discussion of the data against the literature and how it might impact public policies. We have added the following content in the discussion part.

“The results from the GWR model indicate particularly relevant policy implications, as there should be increased recognition of the importance of the context of crime, particularly the location and geography of crime events (Andresen et al., 2021). The emphasis on local differences in the context of crime can bring benefits to specific policy strategies and avoid one-size-fits-all policies. The GWR models present the nonstationary influence between socioeconomic status and crime across London, thus highlighting where more attention is needed. For example, coefficient estimations for the proportion of children differ significantly between central and eastern London. Global models, such as OLS models, which provide only constant coefficient estimations, would suggest a global policy for London as a whole. However, global policy may be highly effective in some areas, while resources may be wasted in others. Andresen et al. (2021) pointed out that the global effect may be undetected, leading to the assumption that the policy is ineffective when in fact it may be highly effective in a small number of places that could benefit greatly from such a policy (Andresen et al., 2021). Our analysis suggests that London should not be considered as a whole to propose global policies but rather that policies should be enacted on a place-by-place basis, depending on the differential spatial impact of the predictors. The division of London is also crucial in policy-making, for example, how many boroughs or districts should be considered as central London. Boroughs can provide rough estimates, but Wards are more detailed and allow for maintaining more detail on spatially varying effects, which is one of the reasons why this analysis is conducted at the Ward level. ”

Besides, we have applied the English language edition by AJE of Research Square.

Round 2

Reviewer 1 Report

The authors state in their response to reviewers that “Additionally, the UK census has only three age groups, including 0-15 years old, 16-64 years old and older than 64 years old. Thus, we only considered the age group between 0 and 15 years old. We hope to conduct further analysis in the future to consider family structure and 15- to 30-year-old populations when data are available. ”

However, I could not find that addition there. Other than that, good job on the revisions.

A final read through before it is sent to proofs would be beneficial.

Author Response

Thanks very much for your comments, we have revised the mistake in the revised manuscript. The reply details can be seen in the attachment file.
